# A Comprehensive Study of De Novo Mutations on the Protein-Protein Interaction Interfaces Provides New Insights into Developmental Delay

**DOI:** 10.3390/biom12111643

**Published:** 2022-11-06

**Authors:** Dhruba Tara Maharjan, Weichen Song, Zhe Liu, Weidi Wang, Wenxiang Cai, Jue Chen, Fei Xu, Weihai Ying, Guan Ning Lin

**Affiliations:** 1Shanghai Mental Health Center, Shanghai Jiao Tong University School of Medicine, School of Biomedical Engineering, Shanghai Jiao Tong University, Shanghai 200240, China; 2Shanghai Key Laboratory of Psychotic Disorders, Shanghai 200030, China; 3State Key Laboratory of Functional Materials for Informatics, Shanghai Institute of Microsystem and Information Technology (SIMIT), Chinese Academy of Sciences, Shanghai 200050, China

**Keywords:** developmental delay, de novo mutation, protein-protein interaction, PPI interface, protein interactome, PsymuKB

## Abstract

Mutations, especially those at the protein-protein interaction (PPI) interface, have been associated with various diseases. Meanwhile, though de novo mutations (DNMs) have been proven important in neuropsychiatric disorders, such as developmental delay (DD), the relationship between PPI interface DNMs and DD has not been well studied. Here we curated developmental delay DNM datasets from the PsyMuKB database and showed that DD patients showed a higher rate and deleteriousness in DNM missense on the PPI interface than sibling control. Next, we identified 302 DD-related PsychiPPIs, defined as PPIs harboring a statistically significant number of DNM missenses at their interface, and 42 DD candidate genes from PsychiPPI. We observed that PsychiPPIs preferentially affected the human protein interactome network hub proteins. When analyzing DD candidate genes using gene ontology and gene spatio-expression, we found that PsychiPPI genes carrying PPI interface mutations, such as *FGFR3* and *ALOX5*, were enriched in development-related pathways and the development of the neocortex, and cerebellar cortex, suggesting their potential involvement in the etiology of DD. Our results demonstrated that DD patients carried an excess burden of PPI-truncating DNM, which could be used to efficiently search for disease-related genes and mutations in large-scale sequencing studies. In conclusion, our comprehensive study indicated the significant role of PPI interface DNMs in developmental delay pathogenicity.

## 1. Introduction

Developmental Delay (DD) is a neurodevelopmental psychiatric disorder that affects individuals’ learning, cognitive, and intellectual abilities from an early age [1]. Several studies have shown that 12–16% of children in America have at least one developmental delay symptom (such as learning disability, speech delay, and sensory impairment) [2,3]. Due to the unclear pathogenesis of developmental delay, one-half of the patients cannot receive the diagnosis on time [4]. Whole-exome sequencing (WES) studies have provided evidence that de novo mutations (DNMs) are strongly associated with the cause of developmental delay [5,6,7,8,9,10,11,12,13,14]. Moreover, mutations that perturb the function and structure of the protein are risk factors for developmental disorders [15,16,17,18]. However, only a rare portion of mutations are linked to the cause of disease. How to effectively locate the pathogenic mutations and relative genes is a question that needs to be answered.

Proteins interact with other proteins to form complexes to perform functions and complete molecular processes in cells [19]. Protein-protein interactions (PPIs) play a crucial role in signal transduction, cell metabolism, and other critical biological processes. The amino acids at the PPI interface determine the specificity and strength of PPIs [20]. Once the mutation alters PPI interface residues, it is likely to perturb the normal interaction and generate disease phenotypes. Such mutations would be identified as disease-related mutations [21]. Research shows that disease-related mutations often prefer to localize at PPI interfaces. [22,23,24,25]. Cheng et al. [22] found that somatic mutations are enriched at the PPI interface in cancer patients, and the mutation-enriched PPIs are highly correlated with the medicine sensitivity and survival period of patients. Yan et al. [26] found that mutations at the SPRED1-NF1 interface can cause Legius syndrome by reducing the strength of PPI. Therefore, studying the PPI interface mutations can provide novel insights to explain disease mechanisms. Nevertheless, such studies are still lacking in the field of neurodevelopmental disease.

Since the PPI interface residue shows a higher evolutionarily conservation [26,27,28,29,30], the pinpoint missense mutation at the PPI interface might have a more deleterious outcome. Previous studies have shown that protein interfaces are enriched in disease-causing missense mutations compared to other protein regions [31,32]. PPI interface DNM missenses have been proven to correlate with the etiology of cancer and autism spectrum disorder [22,26,31,33]. However, the association between PPI interface mutations and developmental delay has not been well-studied. To answer the question, we established a framework to evaluate the harmfulness and pathogenesis of PPI interface DNM missenses in developmental delay.

To investigate the contribution of DNMs missense on the protein-protein interaction interface to developmental delay, we collected DD and sibling controls’ DNM missenses from the PsyMuKB database [34] (Appendix A). Protein-protein interaction interface residue data were collected from the interactome INSIDER database [22]. We first compared the mutation rate and deleteriousness of DD PPI interface DNM missenses with sibling control. Further, we investigated the network properties of proteins that correspond to PPIs that are significantly enriched with PPI interface DNMs missense. Finally, we demonstrated that genes that carry DNM missenses significantly enriched in the PPI interface were associated with disease pathogenesis through gene enrichment analysis. Our study indicated that PPI interface DNMs play a critical role in the pathogenesis of developmental delay. Moreover, our PPI-based framework provides an efficient method to localize disease-related genes and mutations in human neurodevelopmental diseases.

## 2. Materials and Methods

### 2.1. Collection and Preparation of DNM Missenses and PPI Interface Data

We downloaded the de novo mutations data from the PsyMuKB database (http://psymukb.net (accessed on 7 July 2022)) [34] for developmental delay and sibling control subtypes. De novo missense mutations were included in this study. We filtered out frameshift, synonymous, and all non-missense mutations manually. As a result, 4712 in developmental delay and 2074 in sibling control DNM missenses were applied in this study.

We used ANNOVAR (https://doc-openbio.readthedocs.io/projects/annovar/en/latest/ (accessed on 7 July 2022)) to measure the functional impact of DNM missenses by both SIFT and Polyphen-2 scores [35] SIFT predicts whether an amino acid substitution affects protein function based on sequence homology and the physical properties of amino acids. SIFT can be applied to naturally occurring nonsynonymous polymorphisms and laboratory-induced missense mutations [36]. The SIFT score of a mutation ranges from 0–1, and can be classified as deleterious (SIFT ≤ 0.05) and tolerated (SIFT > 0.05). PolyPhen-2 (Polymorphism Phenotyping v2 (http://genetics.bwh.harvard.edu/pph2/ (accessed on 7 July 2022)) is a tool that predicts the possible impact of an amino acid substitution on the structure and function of a human protein using straightforward physical and comparative considerations [37]. The Polyphen-2 score of a mutation ranges from 0–1, and can be classified as probably damaging (≥0.957), possibly damaging (0.453 ≤ polyphen-2 ≤ 0.956), and benign (≤0.452). We obtained SIFT and Polyphen-2 scores from the ANNOVAR annotation database in this analysis. When the DNM missenses’ amino acid change hits the protein interface residue index, we consider such mutations occurring on the PPI interface.

Protein-protein interaction interface data were downloaded from the interactome INSIDER database (http://interactomeinsider.yulab.org/ (accessed on 7 July 2022)) [22]. We only chose the protein-level residue data with the highest confidence interfaces of H. Sapiens from the database. The highest confidence interface included interface residues calculated from PDB structures, homology models, and the “Very High” and “High” interface potential categories from ECLAIR. We used the Uniprot ID mapping tool (http://www.uniprot.org/uploadlists/ (accessed on 7 July 2022)) [38] to convert Uniprot ID to Gene ID.

### 2.2. Enrichment of DNM Missenses on Interaction Interfaces

A total number of 473 in DD and 125 in control proteins with PPI interface DNM missenses containing at least one interaction interface and one known domain was included for calculating DNM missense distribution. The protein sequences were divided into ‘interaction interface’, ’domain’, and ‘other’. Interaction interfaces were determined by the interactome INSIDER database [22]. Domain refers to protein domains that exclude interaction interface in INSIDER. The rest of the sequences were considered as ‘other’. The possibility of mutations occurring in the above three regions *p* was calculated by adding the total sequence length of each region in all proteins and dividing it by the length of all proteins combined. The number of observed mutations in each region was called *S*, and *N* is the total number of dnMis missense mutations. An exact binomial test was computed from *p*, *S*, and *N*. CIs are based on the 95% CI for an exact binomial and then transformed to the risk ratio (enrichment).

### 2.3. Significance Test of PPI Interface Mutations and Identification of PsychiPPI

A pair of protein-protein interactions harboring a statistically significant excess number of PPI interface DNM missenses in one or the other of the two protein-binding partners would be defined as a PsychiPPI. For each gene *g_i_* and respective interfaces, we assumed that the observed mutation number for a given interface follows a binomial distribution, binomial (*T*, *p_gi_*), T is the total mutation number observed in one gene, and *p_gi_* is the estimated mutation rate for the region of interest. Length(*g_i_*) is the sequence length of the protein product of gene *g_i_. p_gi_* = $\frac{length\;of\;interaction\;interface}{length\left(g_{i}\right)}$. For each interface, we computed the *p*-Value—the probability of observing > *k* mutations around this interface out of T total mutations observed in this gene—using the following equation:$$P\left(X\geq k\right)=1-P\left(X<k\right)=1-\overset{k-1}{\underset{x=0}{\sum }}\left(\begin{array}{c} T\\ x \end{array}\right)p_{gi}^{x}{\left(1-p_{gi}\right)}^{T-x}$$

The significance of each PPI was defined as the *p*-Values of two proteins. All *p*-Values were adjusted for multiple testing using the FDR correction. A PPI will be recognized as PsychiPPI only if *q* ≤ 0.05.

### 2.4. Interactome Network Analysis of PsychiPPI Identified DD Candidate Proteins

We downloaded the comprehensive human binary protein-protein interactome data from the interactome INSIDER database (http://interactomeinsider.yulab.org/ (accessed on 7 July 2022)) [22]. The interactome network only included the PPIs, which are experimentally determined. A total amount of 121,575 PPIs (edges or links) connecting 15,046 unique proteins (nodes) were present in the network.

The network was visualized using Cytoscape [39]. Protein’s characteristics within the network, such as degree, betweenness, and the average shortest path length, were analyzed by Network Analyzer, a plugin in Cytoscape.

### 2.5. Haploinsufficiency and Loss of Function Tolerance Analysis of Candidate Genes

The pLI score shows the tolerance of a candidate gene to the loss of function (Lof) on the basis of the number of protein-truncating variants. The pLI score ranges from 0–1 for the most tolerant to the most intolerant genes, respectively. A gene with a pLI score over 0.9 will be classified as intolerant to Lof mutation. We obtained the pLI score for each candidate gene from ExAC database [40].

Human cells have two copies of most genes. If a mutation alters one copy, the other is usually sufficient to maintain gene function. For haploinsufficient genes, both copies must be functioning for the organism to have a normal life. A gene with a haploinsufficiency score over 0.9 will be classified as a haploinsufficient gene. We acquired the haploinsufficiency score for each candidate gene from the ClinGen database [41].

### 2.6. Gene Ontology and Gene Set Analysis

We used Metascape (https://metascape.org/gp/index.html#/main/step1 (accessed on 7 July 2022)) [42] for enrichment analysis of genes in the Gene ontology [43] and DisGeNET [44]. The *p*-Values were calculated by a hypergeometric test.

### 2.7. Spatio-Temporal Expression Patterns of Candidate Genes

The Spatio-temporal expression data of respective genes were downloaded from Human Brain Transcriptome database (https://hbatlas.org/ (accessed on 7 July 2022)) [45]. The expression images displayed log2-transformed signal intensity across analyzed regions/areas and periods using a heat map color scale from low (blue). to high (red).

## 3. Results

### 3.1. More De Novo Missense Mutations Were Observed on the Protein-Protein Interaction Interface in DD Patients Than in Controls

Large-scale studies have reported that disease-related mutations are significantly enriched at the protein-protein interaction interface [22,23,24,25,32,33,46,47]. Those interface mutations can alter the specific interface residue to influence the original PPI, which is crucial in the pathogenesis of many diseases and related genes [20,30,46,48]. To study the relationship between de novo missense mutations and developmental delay, we extracted the DD-related DNM missenses from the PsymuKB database [34]. We then mapped them to the PPI interface residues using interactome INSIDER [22] to investigate the PPI interface mutation rate between DD patients and sibling control. First, in DD, PPI interfaces covered 4.06% of proteins harboring these DNM missenses and 10.10% of the DNM missenses located at interaction interfaces (Enrichment = 2.49, *p*-Value = 2.03 × 10^−70^ by two-tailed exact binomial test). In sibling control, DNM missenses showed a relatively lower enrichment at the PPI interface (observed 6.12% vs. expected 3.49%, Enrichment = 1.75, *p*-Value= 3.80 × 10^−9^) (Figure 1a). Though we observed that both DD patients and sibling controls showed more enrichment missenses on PPI interaction interfaces than non-interface regions, PPI interfaces in DD still showed significantly more missense mutations than in controls. We observed that the rate of missense mutation on PPI interface in DD patients is significantly higher than in sibling control (10.10% vs. 6.12%, 1.65-fold, *p*-Value= 5.92 × 10^−8^ by a two-tailed Fisher’s test) (Figure 1b). Since PPI interface residues determine the strength and specialty of protein interactions, which is critical to the protein function, the significantly higher PPI interface DNM missense rate and enrichment make PPI interface DNM missenses could contribute to DD etiology by altering the PPI interface residue. 

### 3.2. DNM Missenses at PPI Interface Are Significantly More Deleterious in DD Patients Than in Sibling Control

Since the PPI interface residues are more evolutionary conservative than other regions of proteins, theoretically, mutations that occur in such positions could lead to a deleterious result [28,29,30]. To verify the theory and to study the impacts on PPI interface by DD DNM missenses, we used SIFT [36] and PolyPhen-2 [37] predictions to evaluate the possible outcome of both mutations happening on or off the PPI interface between DD and sibling control. We observed that PPI interface mutations were more likely to be classed as ‘deleterious’ and ‘possibly damaging’ in SIFT and PolyPhen-2 results (Appendix A). Moreover, SIFT predictions results showed that DD interface mutations were significantly deleterious than interface mutations from sibling control (81.26% vs. 67.21%, *p*-Value = 0.0013 by a two-tailed Fisher’s test) (Figure 1c). The PolyPhen-2 predictions showed the same result that DD PPI mutations were more likely to damage the protein function than healthy control (71.46% vs. 52.80%, *p*-Value = 1.06 × 10^−4^) (Figure 1d). Moreover, we found that the PPI interface DNM missenses have a higher chance of being deleterious in all of the DD patients’ DNMs (Appendix A). Our SIFT and Polyphen-2 prediction results (Figure 1c,d) showed that DNM missenses at DD PPI interface might have a significantly higher chance of negatively impacting proteins and causing diseases. Additionally, mutations classified as ‘possibly damaging’ tend to disrupt the respective PPI [31], suggesting that PPI interface DNM could associate with the pathogenesis of developmental delay.

### 3.3. Extracting Potential DD Genes and DNM Missenses by Identifying PsychiPPI

Our results showed that DD DNM missenses preferentially occur at the PPI interface with deleterious impacts on the proteins. However, it is unlikely that every PPI interface mutation is pathogenic in DD. Research in other area studies has already shown that PPIs with interaction interface enriched with somatic mutations could be associated with the etiology of the disease [32]. Therefore, we hypothesized that PPIs with interacting interfaces significantly enriched with DNM missense would contribute more to the pathogenesis of DD than PPIs with unaffected interacting interfaces. Thus, we defined PsychiPPI as those PPIs harboring a statistically significant excess number of DNM missenses at the interacting interface (See Methods) to investigate their characteristics. We used a binomial statistical model to calculate the significance of each PsychiPPI. We observed 302 PsychiPPIs among a total of 5562 PPIs, reaching a significance level (*q*-Value ≤ 0.05) after FDR adjustment (Appendix A and Appendix A).

Given the potential severity of PsychiPPIs, we subsequently considered genes carrying PsychiPPI interface mutations as DD candidate genes and identified 42 DD candidate genes from PsychiPPI set to investigate the genes which carry the DNM missenses in DD. We observed that only 24 of 42 have been reported to associate with the etiology of DD, and 18 of them have not been reported yet (Table 1). By visualizing the PsychiPPI network via Cytoscape [39], we observed that some of the candidate proteins’ mutations could directly affect the interaction with several partner proteins by altering the specific residue of the PPI interface (Appendix A). For instance, p.Glu198Lys and p.Pro201Arg in *PPP2R5D* may directly perturb 40 PPI interactions, and the SIFT and Polyphen-2 results both showed above missense mutations are deleterious, suggesting that the mutation impact might disrupt these PPIs. In sum, by defining PsychiPPI and extracting associated proteins, we systematically defined the potential pathogenic genes with DNM missense enriched in PPIs interacting interfaces in developmental delay for the following analyses.

### 3.4. PsychiPPI Affected Hub Proteins in the Human Interactome

Studies in various inherited diseases show that protein networks encoded by disease-associated genes have distinct interactome network properties, such as degree, betweenness, and average shortest path length, compared to networks encoded by non-disease-associated proteins [70,71]. Previous research reported that disease-associated genes tend to encode hub proteins which mediate a significantly larger number of protein interactions than other proteins in the human protein interactome [72,73]. To investigate if PsychiPPI proteins carrying mutations from DD have a greater impact on human interactome than all proteins carrying PPI interface mutations in DD and sibling control. We analyzed the candidate proteins’ topological network features from an unbiased human protein-protein interactome network constructed by PPIs generated from experiments only (see Methods).

First, we calculated the degree of all proteins from PPI with an interacting interface carrying DNM missenses. We demonstrated that candidate proteins identified by PsychiPPI, on average, have a significantly higher degree than all DD proteins harboring interacting interface DNM missenses (mean ± s.e.m.: 28.76 ± 1.61 vs. 22.83 ± 0.69, fold change (FC) = 1.26, *p*-Value = 9.83 × 10^−4^ by a two-tailed U-test), and PPI interface mutated proteins from sibling control (mean ± s.e.m.: 28.76 ± 1.61 vs. 18.58 ± 1.34, FC = 1.55, *p*-Value = 1.64 × 10^−6^) (Figure 2a). We further analyzed the same group of proteins by calculating their betweenness in the network. We found that DD candidate proteins also have a significantly higher betweenness value than all the PPI interface mutated proteins in DD patients (mean ± s.e.m.: 0.075 ± 0.004 vs. 0.061 ± 0.002, FC = 1.23, *p*-Value = 0.012) and sibling control (mean ± s.e.m.: 0.075 ± 1.61 vs. 0.052 ± 0.004, FC = 1.43, *p*-Value = 5.65 × 10^−5^) (Figure 2b). At last, we explored whether DD candidate proteins identified by PsychiPPIs tend to form interconnected modules within the interactome network. We found that DD candidate proteins identified by PsychiPPI have a significantly shorter average shortest path length compared to all the proteins of PPI with interface missense mutations from DD patients (mean ± s.e.m.: 3.20 ± 0.03 vs. 3.27 ± 0.02, FC = 0.94, *p*-Value = 0.006) or sibling control (mean ± s.e.m.: 3.20 ± 0.03 vs. 3.39 ± 0.04, FC = 0.98, *p*-Value = 3.69 × 10^−5^) (Figure 2c). The result implied that proteins harboring mutations residing on PsychiPPI interfaces preferred to be closely connected in the interactome network. Such inner-connected modules may serve specific roles in the etiology of DD.

In conclusion, by analyzing the topological network properties of selected protein groups, we demonstrated that DD candidate proteins tend to locate at the human interactome network hub, which significantly impacts the protein interactome more than all the other interface mutated proteins. The network analysis can assist in interpreting how PPI interface DNM missenses could affect protein complexes and functional modules.

### 3.5. PsychiPPI-Related DD Candidate Genes Tend to Be Haploinsufficient

It is known that heterozygous deleterious DNM missenses could only affect one copy of the genes, whereas a single copy of the wild-type gene is insufficient to carry out the normal function for haploinsufficient genes [74]. In DD patients included in the study, we observed that genes carrying DNM missense in PsychiPPI exhibited a significantly higher haploinsufficiency score [75] than all the genes carried PPI interface mutation (mean ± s.e.m.: 0.67 ± 0.03 vs. 0.55 ± 0.02, FC = 1.21, *p*-Value = 4.25 × 10^−4^ by a two-tailed U-test). In addition, we found that in the DD group, both kind of genes residing on or off PPI interface mutations had a significantly higher probability of being haploinsufficient (Appendix A). Meanwhile, we demonstrated, in the same group of genes, DD candidate genes were less tolerant to genetic variation. PsychiPPI-related DD candidate genes had an average higher pLI score than all the genes that carry PPI interface mutations in DD (mean ± s.e.m.: 0.85 ± 0.03 vs. 0.64 ± 0.02, FC = 1.33, *p*-Value = 2.69 × 10^−7^), at the same time, DD patients’ dnMis genes showed a higher pLI score [76] in both on or off PPI interface mutated genes than sibling control (Appendix A). Both results from haploinsufficiency and pLI showed DD candidate genes might contribute significantly toward the phenotype of developmental delay via dosage effect [76,77].

### 3.6. PsychiPPI-Related DD Candidate Genes Are Enriched in Curated DD-Related Geneset

Given results from interactome and haploinsufficiency showing the potential pathogenic contribution of DD candidate genes and respective mutations, we next investigated the possible association between the curated disease gene set and candidate genes to evaluate the fundamental connection between genes and disease. Thus, we prioritized all 42 DD candidate genes according to 11 independent sources of evidence with a background gene list as all protein-coding genes. We found that PsychiPPI identified DD candidate genes were significantly enriched in DD-related curated pathogenic gene sets, such as developmental delay gene set (Count: 19, *p*-Value < 0.001), Constraint gene set (Count: 17, *p*-Value < 0.001), FMRP target gene set (Count: 7, *p*-Value = 1.58 × 10^−3^) (Figure 3b), we also observed *SMAD3*, a candidate gene with limited literature studies, is also included in the Developmental delay gene set (Figure 3a).

We also used the DisGeNET database [44] to evaluate the gene-disease association to verify our result with all protein-coding genes as a background gene list. The outcome showed DD candidate genes were enriched in developmental disabilities (Count: 18, log10(*P*) = −23), neurodevelopmental disorders (Count: 16, log10(*P*) = −17), Mental retardation (Count: 15, log10(*P*) = −16, and other diseases with similar symptoms (Figure 3c). The gene set enrichment analysis results indicated the significant association between PsychiPPI-identified DD candidate genes and developmental delay. It also implicated the PPI interface mutations with the cause of DD.

### 3.7. Identification of New Potential DD Genes and De Novo Mutations

The integrative analysis and literature search have proven the association between disease and PsychiPPI-identified DD candidate genes. We observed that 25 genes had been reported to be associated with the pathology of DD out of the total number of 42 candidate genes. In order to find out the potential DD-related genes and mutations from the rest of the unreported candidate genes, we analyzed the enrichment of DD candidate genes in Gene Ontology terms. We found DD candidate genes are significantly enriched in DD-related GO terms, such as cell cycle process (Count: 9, log10(*P*) = −6.21), MAPK cascade (Count: 7, log10(*P*) = −4.38), and skeletal system development (Count: 7, log10(*P*) = −5.24) [78,79] (Figure 4a). We next investigated the presence of unreported potential DD-related genes in GO biological process and cellular component terms. We observed that *FRFR3* is present in developmental maturation, developmental growth, regulation of growth, regulation of MAPK cascade, and response to growth factors, which are highly disease-related biological process terms [80]. Meanwhile, *ALOX5* is a crucial part of cellular components such as the nuclear matrix, ficolin-1-rich granule, and nuclear membrane (Figure 4b). These outcomes indicated the potential relationship between *FGFR3*, *ALOX5*, and developmental delay.

The main symptoms of DD include delayed physical and cognitive development in children before they reach the age of adolescence [81]. These characteristics make spatial brain development in different parts of regions crucial to the mechanism of DD. To evaluate the association between our potential disease-related genes and DD, we investigated the expression trajectories of *ALOX5* and *FGFR3* in neurodevelopmental processes in respective brain regions from the Human Brain Transcriptome database [45]. We observed that the expression of *FGFR3* shows a dramatic rising from the late fetal developmental stage till the end of the late infancy stage in the neocortex, which is highly correlated with the visual and motor cortex in the neocortex [82] (Figure 5a). *ALOX5* also demonstrated a rapid increase in expression from the infancy stage till the end of adolescence in the cerebellar cortex, which controls the physical motors [83] (Figure 5b). Previous studies have shown that the abnormalities in the neocortex and cerebellar cortex could contribute to the phenotype and pathogenesis of developmental delay [84,85,86,87]. 

Moreover, *FGFR3* plays a crucial role in regulating cell growth and neural development [88]. Mutations on *FGFR3* could prematurely trigger intracellular signaling to terminate cell proliferation [89]. *ALOX5* is also related to the progression of neurodegenerative diseases such as Alzheimer’s disease [90]. The suppression of the expression of *ALOX5* could reduce inflammation-induced neuro cell death [91,92]. All results above suggested the potential association of DD-related pathogenesis between *FGFR3* and *ALOX5*.and their respective DNM missenses.

## 4. Discussion

The growing number of research has shown that mutations at the protein-protein interaction interface are highly associated with the etiology of cancer and rare diseases [31,32,33]. Based on these findings, we designed a framework to effectively identify potential disease-related genes and mutations based on the location and enrichment of mutations on the PPI interface. In the present study, we focused on DNM missenses from the PsyMuKB database because of its adequate DD and sibling control de novo mutations data. Our results showed that DD-related DNM missenses are more likely to occur on the PPI interface than in healthy control. Additionally, the deleterious predicted outcome of DD PPI interface DNM missenses proved that such mutations tend to alter protein functions and structures. Using a binomial statistical model, we identified 302 PsychiPPIs harboring a statistically significant excess number of DNM missenses at the PPI interface (PsychiPPI). We extracted 42 DD candidate genes according to the mutated genes in the PsychiPPI. The candidate proteins identified by PsychiPPI preferentially hit the hub protein in the interactome, a common feature for disease-related genes. The Integrative analysis and literature research demonstrated a significant association between DD candidate genes and the etiology of developmental delay, reinforcing the efficiency of our disease-related gene exploring framework. Combining the Gene ontology analysis results and Human Brain Transcriptome database, we showed that *FGFR3* and *ALOX5* could serve as novel developmental delay candidate genes. In sum, our finding suggests the significant role of DNM missenses at PPI interfaces in the pathogenesis of developmental delay. Meanwhile, the evaluation of DD-related PsychiPPI could contribute to discovering new disease-related genes and respective etiology.

The protein-protein interface determines the specificity and strength of protein interactions, mutations on the PPI interface residue could disrupt, decrease or even increase the normal PPI, which could cause disease by disturbing the function of proteins [20,21]. However, not all genes with PPI interface mutation are pathogenic. Our study shows that most genes with a statistically significant excess number of DNM missenses at the PPI interface (PsychiPPI) in DD have been reported to be disease-related. The result is innovative in neuropsychiatric disease but less surprising in other diseases like cancer. The previous study has found that PPI harboring a significant excess number of somatic mutations at the PPI interface is significantly associated with poor survival rate and drug sensitivity in cancer patients. Our interactome analysis shows that proteins identified by PsychiPPI are preferentially located at the more central position in the whole human protein interactome. Interface mutations could not only disrupt the normal PPI, but it has a chance to cause an unknown consequence by making mutated proteins interact with new partners. Deleterious mutations on those proteins could significantly impact the whole network and generate disease phenotypes. On the other hand, studying PPI network perturbations altered by interface mutations could provide us with a fundamental pathogenic molecular mechanism of widespread disease.

Our result is not only applicable in the exploration of DD-related genes, but it is also suitable for neurodevelopmental diseases such as intellectual disability (ID). We found the same level of enrichment in ID DNM missense on the PPI interface (Appendix A). However, in the result of autism spectrum disorder (ASD) and schizophrenia, we have not acquired a significant difference between sibling controls. A previous ASD study has reported a substantial difference in the enrichment of DNM missenses at the PPI interface [31], which contradicts our result. One possible explanation is that a different interactome interface database was used in our study. The negative result from schizophrenia agreed with previous research that schizophrenia has a weaker de novo signal than other psychiatric diseases [93]. Considering these findings, the level of DNM missense enrichment on the PPI interface must be regarded when our framework is applied to other neurodevelopmental diseases.

There are a few limitations in our current research. For simplicity, our null model for calculating PsychiPPI does not consider the DNA sequence, which can influence the computation of the frequency of nonsynonymous mutations. It is possible that taking into account the nucleotide frequency may improve the predictions, but we expect that the improvement is low. The null model does not consider the known fact that the selective effect of mutations depends on the protein region where it occurs. For instance, mutations in the protein core are more detrimental than mutations in the surface, where the protein-protein interface tends to be, and have higher chances of being eliminated by natural selection at the developmental stage. This might explain our result that mutations at the protein-protein interface happen more frequently than expected under the null model both in the cases and in control, and it is a possible direction to improve the null model that we leave for future work. Previous studies have shown that disease-causing missense mutations are enriched in the protein interface compared to other protein surface regions [31]. However, we recognize that the PPI interface DNMs may be less disruptive than those mutations on protein cores. If the core of proteins is mutated, it might also disturb normal PPI. The INSIDER interactome database was the source of PPI interface data. The majority of the PPI interface data were calculated by deep learning. Despite its sufficient data, the prediction accuracy needs to be modified. One shortage of our study is that we only consider the prediction with high confidence, making the interactome seem incomplete in the study. We believe that the fast-growing interest in determining potential PPI interface residue could provide us with complete interactome data in the future. Although the Gene Ontology and the data from Human Brain Transcriptome database support that *FGFR3* and *ALOX5* and their mutations are possible diseases related. Moreover, previous studies have reported that mutated postmitotic principal neurons *FGFR3* could result in cortical dysplasia and axonal tract abnormality during brain development in mice [94]. Moreover, Other studies have found that the downregulation of *FGFR3* signaling may be essential for maintaining the balance of neurogenesis and neuronal differentiation during cortical development. Several *FGFR3* de novo mutations were identified in thanatophoric dysplasia (TD) patients. Some patients with these mutations have an intellectual disability and severe skeletal deformities, which correspond to the symptom of DD [95]. Additionally, research has shown that *ALOX5* is related to memory deficits and synaptic dysfunction in a mouse model of Alzheimer’s disease [96]. Despite the supportive evidence, our hypothesis is not entirely settled. Studies involved in cell or animal models are required to prove our findings.

This study showed a significant association between PPI interface DNMs and developmental delay, combined with our PsychiPPI-based disease-related genes searching framework. With the development of whole exome sequencing, researchers worldwide could identify thousands of DNM missenses in a large-scale study within a relevantly short time. We may contribute to discovering disease mechanisms and early diagnosis of developmental delay and other potential neuropsychiatric diseases.

## Figures and Tables

**Figure 1 biomolecules-12-01643-f001:**
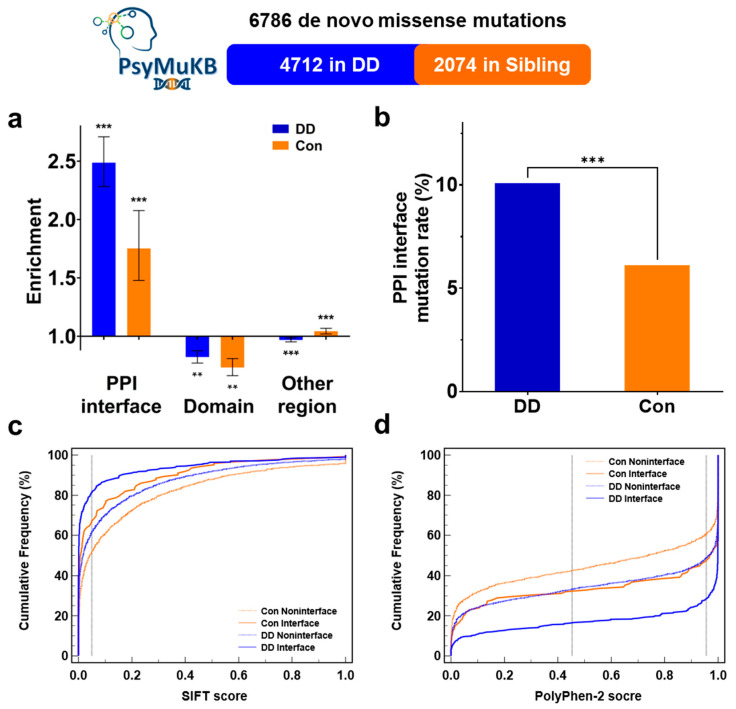
PPI interface DNMs enrichment and deleteriousness in DD. (**a**) Distribution of DNMs from the PsyMuKB across different protein locations. Enrichment was calculated as the ratio of the mutation rate of DNMs at the protein-protein interaction interface over the possibility of interface residues on corresponding proteins. *p*-Values were calculated using a two-tailed exact binomial test (** *p* < 0.01; *** *p* < 0.001). The error bars indicate standard error. (**b**) Mutations rate of DNM missenses at protein-protein interaction interface between developmental delay and sibling controls from the PsyMuKB. *p*-Values were calculated by a two-tailed Fisher’s test. (**c**,**d**) Cumulative frequencies of SIFT (**c**) and PolyPhen-2 (**d**) scores for protein-protein interface mutations.

**Figure 2 biomolecules-12-01643-f002:**
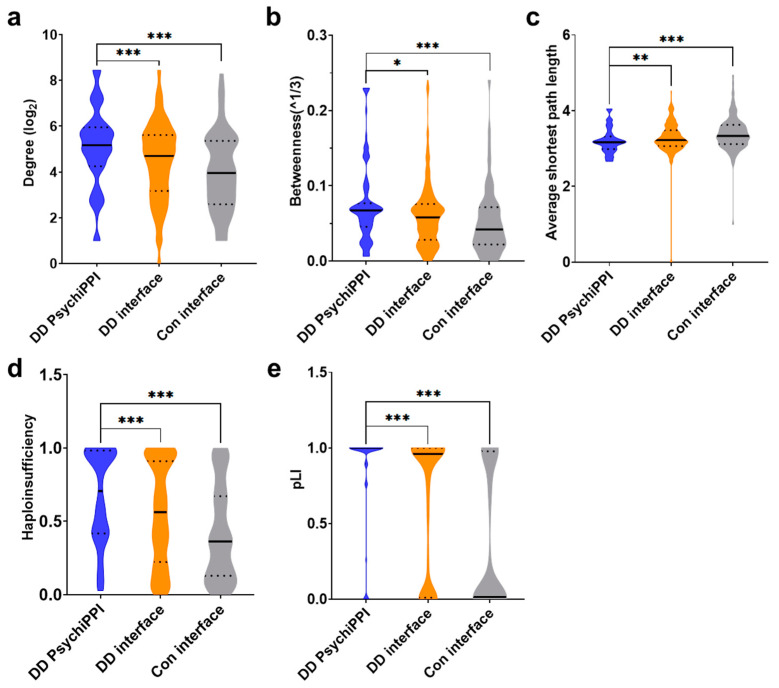
DD candidate genes’ human interactome and haploinsufficiency analysis. (**a**,**b**), Degree (**a**) and betweenness (**b**) distributions of DD candidate proteins identified by PsychiPPI, and all proteins carry PPI interface mutations in DD patients and sibling controls. Degree and betweenness values are transformed by log2 and cube root (^1/3) for presentation purposes, respectively. (**c**), Average shortest path length distributions of proteins in the respective group.(**d**,**e**), Haploinsufficiency (**d**) and pLI (**e**) distributions of genes carry PPI interface DNM missenses. Genes with available haploinsufficiency or pLI scores were included in the analysis. (DD PsychiPPI, *n* = 113; DD interface, *n* = 472; Con, *n* = 122) Violin plots: black horizontal solid line, median; black dotted line, interquartile range; whiskers, upper and lower limits; the width of each plot is proportional to element abundance, *p*-Values were calculated using a two-tailed *U*-test (* *p* < 0.05; ** *p* < 0.01; *** *p* < 0.001).

**Figure 3 biomolecules-12-01643-f003:**
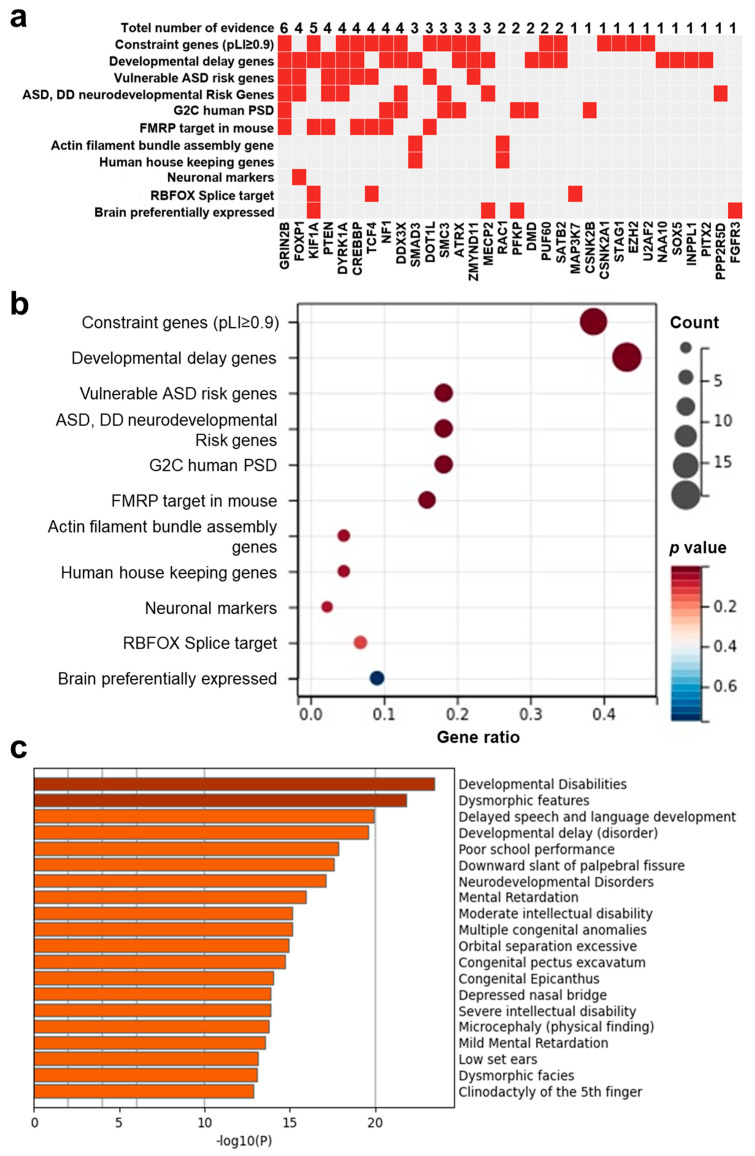
The pathogenicity of DD candidate genes was identified in the present study. (**a**,**b**), Integrative analysis of the potential pathogenicity of DD candidate genes (**a**), DD-related pathogenesis gene set enrichment analysis of candidate genes (**b**), and eleven independent sources of evidence were used to analyze DD candidate genes. Gene ratio is the number of differential genes enriched in a particular gene set to the total number of candidate genes. (**c**), DisGnNET enrichment analysis of DD candidate genes.

**Figure 4 biomolecules-12-01643-f004:**
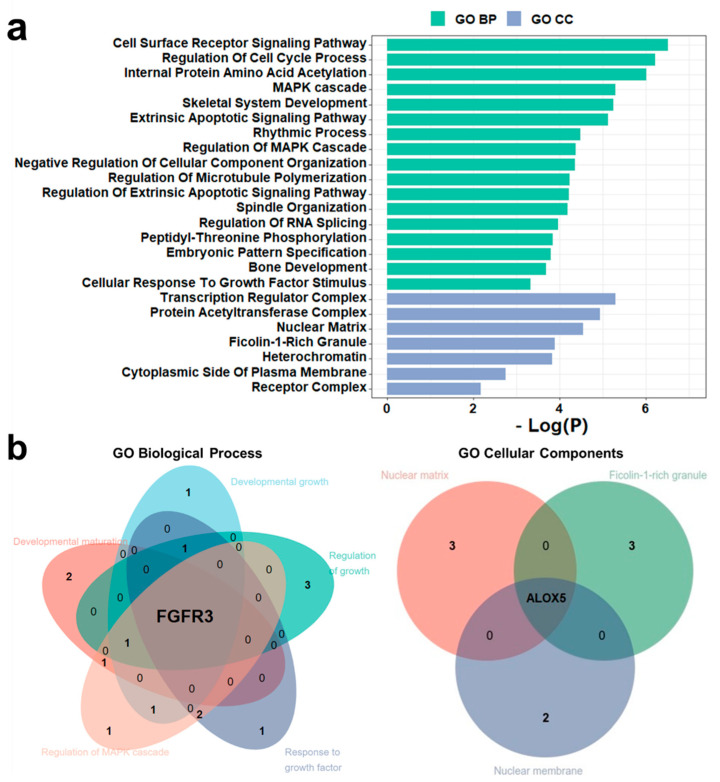
*FGFR3* and *ALOX5* are potential DD-related genes according to GO enrichment analysis. (**a**), Gene ontology enrichment analysis of DD candidate genes. (**b**), *FGFR3* is present in DD-related GO BP sub-term: developmental maturation, developmental growth, regulation of growth, regulation of MAPK cascade, and response to growth factor. *ALOX5* is present in DD-related GO CC sub-term: nuclear matrix, ficolin-1-rich granule, and nuclear membrane.

**Figure 5 biomolecules-12-01643-f005:**
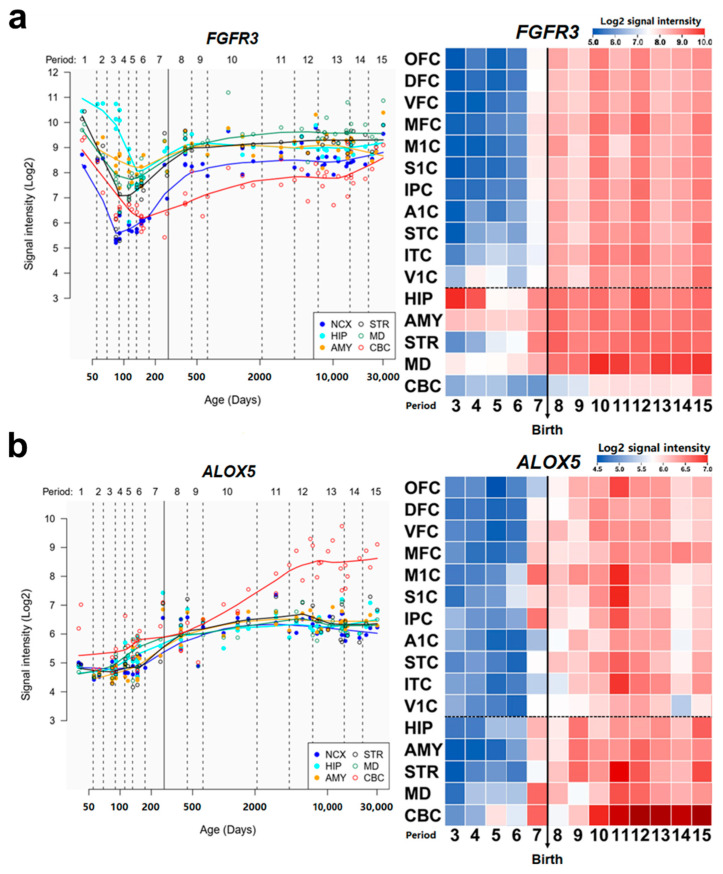
Spatio-temporal expression patterns of *ALOX5* and *FGFR3*. (**a**,**b**), left: spatio-temporal expression pattern of *ALOX5* (**a**) and *FGFR3* (**b**) across age. right: heat map matrix representations of spatio-temporal expression of *ALOX5* and *FGFR3*. The heatmap display log2-transformed signal intensity across analyzed regions and periods using a heat map color scale from low (blue) to high (red). The dashed horizontal dotted line separates NCX areas from other brain regions. The birth time is marked by a vertical solid line. Abbreviation: NCX, Neocortex; OFC, Orbital prefrontal cortex; DFC, Dorsolateral prefrontal cortex; VFC, Ventrolateral prefrontal cortex; MFC, Medial prefrontal cortex; M1C, Primary motor cortex; S1C, Primary somatosensory cortex; IPC, Posterior inferior parietal cortex; A1C, Primary auditory cortex; STC, Posterior superior temporal cortex; ITC, Inferior temporal cortex; VIC, Primary visual cortex; HIP, Hippocampus; AMY, Amygdala; STR, Striatum; MD, Mediodorsal nucleus of the thalamus; CBC, Cerebellar cortex.

**Table 1 biomolecules-12-01643-t001:** List of all DD candidate genes identified by PsychiPPI.

Gene	Interface/ Mutation	Protein Change			SIFT	Polyphen2	Ref
*CREBBP*	6/10	p.C1788W p.H1791D	p.C1370W p.N1551K	p.Y613C p.Y651C	D	D	[49]
*PPP2R5D*	14/16	p.E198K	p.P201R		D	D, P	[50]
*PUF60*	3/3	p.E138K	p.D142N	p.G448E	D, T	D, B	[51]
*KIF1A*	4/8	p.R307P	p.R465W	p.R307Q	D	D	[52]
*NAA10*	4/5	p.R82Q p.H114P	p.H120P	p.R83C	D	D, P	[13]
*ZMYND11*	5/6	p.R546W p.C520R	p.C489R p.R583W	p.V493I	D, T	D, B	[53]
*DDX3X*	8/15	p.C325R p.R360C p.A217V	p.P568L p.C452Y	p.V535I p.R475G	D	P, D, B	[54]
*CSNK2A1*	6/8	p.K62R p.I38M	p.F61I	p.R191Q	D	D, P	[14]
*MECP2*	6/9	p.R145C	p.R133H	p.T158M	D	D	[55]
*ATRX*	1/2	p.L191F			D	D	[56]
*DYRK1A*	5/7	p.L169P p.S337P	p.L198P p.A277P	p.D287V	D	D	[57]
*FOXP1*	5/7	p.A534E,	p.F500L	p.A533E	D	D	[58]
*SOX5*	2/2	p.A561P	p.A596P		D	D	[59]
*TCF4*	2/4	p.R576W	p.R579W		D	D	[60]
*CSNK2B*	1/2	p.R86C			D	D	[14]
*SATB2*	5/5	p.E402K p.R389L	p.R389C p.R399H	p.G515S	D	D, P	[61]
*SMC3*	3/4	p.G1188A	p.Q1147E		D	D	[62]
*GRIN2B*	3/8	p.E807K	p.S628F	p.N615K	NA	D	[63]
*DMD*	1/1	p.D97N			D	B	[64]
*U2AF2*	1/4	p.T252I			D	D	[65]
*RAC1*	2/3	p.Y64D	p.P73L		D	P, D	[66]
*PTEN*	2/2	p.L313F	p.D268E		D, T	D, B	[67]
*EZH2*	2/2	p.R679C	p.R679H		D	D	[68]
*NF1*	1/2	p.R1830C			D	D	[69]
*STAG1*	2/2	p.R216G			D	D	NA
*PTPN3*	1/1	p.K708N			D	D	NA
*PITX2*	1/1	p.R115G			D	D	NA
*MAP3K7*	1/2	p.R238Q			D, T	B	NA
*FGFR3*	1/1	p.G380R			D	D	NA
*TOR1AIP1*	1/1	p.R438H			D	B	NA
*ING4*	1/1	p.Y195C			D	D	NA
*NFAT5*	1/1	p.E462D			D	D	NA
*DOT1L*	1/1	p.R292C			D	B	NA
*INPPL1*	1/2	p.R581Q			D	D	NA
*PFKP*	2/2	p.I473V	p.I651V		D	B	NA
*RBM12*	1/1	p.V867M			D	D	NA
*FGFR4*	2/2	p.D507N	p.R567G		D	D, P	NA
*ALOX5*	1/1	p.G528S			D	D	NA
*SMAD3*	1/1	p.G249C			D	D	NA
*TDRD7*	1/1	p.T6I			D	D	NA
*ANKRD28*	1/1	p.G247E			D	D	NA
*H2BC3*	1/2	p.S56L			NA	B	NA

D, damaging, T, tolerate, B, benign. P, probably damaging. NA, no association reported yet.

## Data Availability

https://github.com/Magiciantype69/PsychiPPI (accessed on 7 July 2022).

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
