# Peer review of "A Comprehensive Study of De Novo Mutations on the Protein-Protein Interaction Interfaces Provides New Insights into Developmental Delay"

_biomolecules, 2022, doi:10.3390/biom12111643_

Round 1

Reviewer 1 Report (Previous Reviewer 3)

The authors made minimal changes to mention the limitations of their null model; they are minimal and not clear enough but, since it is already the third revision, I agree that the paper can be published, provided that the authors state the limitations of the null model in a less hidden form. These are my suggestions to present a clearer discussion of the limitations of the null model:

"Due to the different frequencies of non-synonymous between somatic and de novo mutations on the nucleotide sequence, our null hypothesis does not take into account the frequency of non-synonymous DNMs on nucleotide sequence while calculating PsychiPPI."
->
"For simplicity, our null model for calculating PsychiPPI does not take into account the DNA sequence, which can influence the computation of the frequency of non-synonymous mutations. It is possible that taking into account the nucleotide  frequency may improve the predictions, but we expect that the improvement is low."

I do not agree with the answer of the Authors that the De Novo Mutations do not pass the selection filter. If the mutation compromises some stage of the development and does not result in a viable embryo it will be eliminated by natural selection. Moreover, their sentence in the paper is not clear, and I suggest that they modify to express a concept similar to the one proposed below:

"The null model we used may neglect the potential impact of mutations occurring on other regions of proteins, such as protein cores."
->
"The null model does not consider the known fact that the selective effect of mutations depends on the protein region where it occurs. For instance, mutations in the protein core are more detrimental than mutations in the surface, where protein-protein interface tend to be, and have higher chances of being eliminated by natural selection at the developmental stage. This might explain our result that mutations at the protein-protein interface happen more frequently than expected under the null model both in the cases and in the control, and it is a possible direction to improve the null model that we leave for future work."

Of course the authors are free to formulate the caveat as they prefer, but they should express it in a way that the readers may be aware of it.

Author Response

Reviewer 2 Report (Previous Reviewer 2)

I do not have further comments for the authors.

Author Response

We thank the reviewer for the positive support and all the valuable suggestions on our manuscript. We have revised our research with the previous comments provided by the reviewer.

This manuscript is a resubmission of an earlier submission. The following is a list of the peer review reports and author responses from that submission.

Round 1

Reviewer 1 Report

This study showed a significant association between PPI interface DNMs and DD, combined with a PsychiPPI-based disease-related genes searching framework. 

This work is limited by the use of a single interactor module, a limited use of PsychiPPI, and the use of too few prediction softwares. Using combination of SIFT and PolyPhen2 only is not actual standard in WES studies.

Reviewer 2 Report

This manuscript by Maharjan et al. brings new evidence about the impact that protein-protein interaction (PPI) interface exerts on missense de novo mutations (DNMs) associated with neuropsychiatric disorders, such as developmental delay (DD).

Using PsyMuKB database the Authors show that DD patients exhibit a higher rate of harmful de novo missense mutations on the PPI interface than sibling control. By analyzing DD candidate genes using gene ontology and gene spatio-expression, the Authors find that PsychiPPI genes carrying PPI interface mutations are enriched in development-related pathways and the development of the neocortex, and cerebellar cortex, suggesting the potential involvement of FGFR3 and ALOX5 in the pathogenesis of DD.

Analysis of the data and conclusions are correct but they appear highly dependent on the database source used since some of the conclusions about the enrichment in ID DNM missense on the PPI interface were not significant. Also in the analysis of autism spectrum disorder (ASD) and schizophrenia, the Authors have not found a significant difference between sibling controls while a previous ASD study has reported a substantial difference. 

I do not have major criticisms, but I invite the Authors to clarify better the possible impacts their study may have on the diagnosis and care of genetic mutations leading to DD.

1.      What kind of advance the present findings bring in the design of new therapies for treating DD?

2.      There is no specific indications on how the different affected genes may affect synaptic transmission and neuronal development.

3.      How the two indicated mutated genes (FGFR3 and ALOX5) are able to alter neuronal excitability, synapse maturation and neuronal development?

4.      I am quite surprised that no neuronal ion channel de novo missense mutation is involved in DD and ASD.

Reviewer 3 Report

This paper reports a nice computational study that compares de novo missense mutations (DNM) in the PsyMuKB database between developmental delay (DD) patients and sibling healthy controls. They find that DNM mutations tend to be located on known protein-protein interaction interfaces (PPI) significantly more in DD patients than in controls and that DD-related DNM mutations at PPI are predicted to be more pathogenic according to the SIFT and PolyPhen scores. These results are not surprising, given that PPI are key regulators of cellular processes, but they are interesting.

The authors also characterize a set of 42 genes that they call PsychiPPI, in which the number of DNM at the PPI is significantly higher than expected according to a null model developed by the authors. 25 of these genes are already known to be mechanistically associated with DD but 17 are novel. Comparing PsychiPPI with other DNM at PPI in DD patients that are not significant and with healthy controls shows, the authors find that PsychiPPI tend to occur in proteins with higher interaction degree and betweenness, and they tend to be more haploinsufficient and loss-of-function intolerant, which suggests that dosage effects may play a role in DD. These genes are often present in curated pathogenic gene sets. Two of the novel identified genes were further submitted to GO term analysis and expression analysis in time and space, which supports their possible involvement in developmental delay, although no causal mechanism is proposed from this analysis.

I like the paper, but I have some concerns on the null model that is presented at line 122. This model assumes that all amino acids are equally likely to be mutated. However, the healthy controls significantly violate the model (l.162: we observed that both DD patients and sibling controls showed more missenses on PPI interaction interfaces than non-interface regions), which in my opinion suggests that the model is not accurate. There are two important factors that the null model does not take into account.
First, the nucleotide sequence and the genetic code. Ideally, for computing the null probability, the gene sequence should be considered and the genetic code should be used for computing the probability that there are k missense mutations at the interface. It would be possible to use experimental data on the known mutations frequencies from any nucleotide to any other (they are known in the human genome) or to assume for simplicity that these frequencies are equal for all nucleotide pairs, but at least the effect of the genetic code should be considered.
Secondly, the mutations that are tolerated by natural selection are known to strongly depend on the 3D structure of the protein such that mutations at buried positions are more strictly selected against because they tend to exert more deleterious effects on protein structure and folding stability. This effect may be relevant, because PPI are located on the surface of the protein and this may contribute to the observation that PPI mutations are more frequent than expected under the null model also on the control set. To improve the null model it would be advisable to determine if a site is on the core, surface or intermediate and treat the three classes separately. This could be done at least for proteins of known structure, but the improvements in protein structure prediction makes this feasible also for other proteins.

Furthermore, even if it is plausible that mutations at PPI produce more severe consequences than at other sites, it is of course conceivable that also other mutations play a role in DD. If also not PPI mutations are identified, the role of PPI in DD would be a result of this study; as the study is conceived now, I see it more as a hypothesis than as a result. I think that this aspect should be discussed. To improve the study, I'd like to suggest the following. In the presented work, the null model is parameterized on mutations that are occur outside the PPI and its significance is tested at the PPI. Wouldn't it be possible to parameterize the null model on the control set and the global number of mutations of the DD set, and test it for each protein of the DD set, separately for PPI and not PPI mutations? In this way to determine if any specific protein significantly contributes to DD even if the mutations do not occur at the PPI, and the relative role of PPI versus non-PPI mutations could be more fairly assessed.

Minor comments:

l.16: DMN -> DNM
l.145: Please remove the paragraph "This section may be divided by subheadings..."
Figure 1: What is "Other"? (No interface and not domain). The error bars do not seem to support the very high significance. How are they exactly computed? Why aren't they shown in Fig.1b?
L.185: Please explain on what the SIFT and PolyPhen predictions are based.
L.288: Please define and explain pLI.
L.300: "We found that PsychiPPI identified DD candidate genes were significantly enriched in DD-related curated pathogenic gene sets". I may have lost something, but isn't this obvious? After all, the PsychiPPI genes were found in a database of mutations expressed in DD patients, and the curators of pathogenic gene sets look into these kinds of databases to identify pathogenic genes... Please explain why the "pathogenic gene sets" constitute independent evidences with respect to PsyMuKB.

Round 2

Reviewer 1 Report

The manuscript underwent minimal changes

Reviewer 3 Report

I feel that the authors did not consider my main concern on their null model. I understand that, at this stage, modifying the null model and redo the computation would be too much effort and time, but I think that they should at least recognize its main limitations. Which are not due to the limitations of the PPI interface database, as they write in the response and in the revised version, but have more fundamental grounds: (1) The null model does not use the nucleotide sequence in order to compute the frequency of non-synonymous mutations (please correct me if I am wrong, and explain clearly in the paper). (2) The null model does not take into account that surface mutations are much less  disruptive and have a higher probability to pass even the somatic selection filter, which is quite relevant here, since the interface mutations that the authors assess happen at the protein surface.